# Glia-Neurotrophic Factor Relationships: Possible Role in Pathobiology of Neuroinflammation-Related Brain Disorders

**DOI:** 10.3390/ijms24076321

**Published:** 2023-03-28

**Authors:** Ewelina Palasz, Anna Wilkaniec, Luiza Stanaszek, Anna Andrzejewska, Agata Adamczyk

**Affiliations:** 1Mossakowski Medical Research Institute, Polish Academy of Sciences, 02-106 Warsaw, Poland; 2Center for Advanced Imaging Research, Department of Diagnostic Radiology and Nuclear Medicine, University of Maryland, Baltimore, MD 21201, USA

**Keywords:** neurotrophic factors, neuroinflammation, mitochondrial dysfunction, glial cells, oxidative stress, neuroprotection

## Abstract

Neurotrophic factors (NTFs) play an important role in maintaining homeostasis of the central nervous system (CNS) by regulating the survival, differentiation, maturation, and development of neurons and by participating in the regeneration of damaged tissues. Disturbances in the level and functioning of NTFs can lead to many diseases of the nervous system, including degenerative diseases, mental diseases, and neurodevelopmental disorders. Each CNS disease is characterized by a unique pathomechanism, however, the involvement of certain processes in its etiology is common, such as neuroinflammation, dysregulation of NTFs levels, or mitochondrial dysfunction. It has been shown that NTFs can control the activation of glial cells by directing them toward a neuroprotective and anti-inflammatory phenotype and activating signaling pathways responsible for neuronal survival. In this review, our goal is to outline the current state of knowledge about the processes affected by NTFs, the crosstalk between NTFs, mitochondria, and the nervous and immune systems, leading to the inhibition of neuroinflammation and oxidative stress, and thus the inhibition of the development and progression of CNS disorders.

## 1. Introduction

Neuroinflammation is defined as an inflammatory response within the central nervous system (CNS) and is mediated by resident CNS glial cells. Neuroinflammation is often considered a pathological process, but it should be remembered that its destructive role is associated not so much with presence as with intensity and dynamics [1]. The first line of defense is microglia, which recognize the pathogen, recruit immune system cells, remove the pathogen, and repair damaged tissue. Microglia become activated by pathogen-associated molecular pattern molecules (PAMPs) that are derived from microorganisms, and by damage-associated molecular pattern molecules (DAMPs) that are cell-derived and initiate and perpetuate immunity in response to trauma, ischemia, and tissue damage, either in the absence or presence of pathogenic infection (Figure 1) [2]. Activated microglia express various surface molecules, such as Fc receptor, integrins (CD11b, CD11c), CD14, major histocompatibility complex (MHC) molecules, toll-like receptors (TLRs), scavenger receptors, and cytokine/chemokine receptors [3], and may release mediators such as cytokines, matrix metalloproteinases (MMP), reactive oxygen species (ROS), nitric oxide (NO), glutamate, and neurotrophic factors (NTFs) [4]. Microglia can be categorized into two opposite types: classical (M1), which release inflammatory mediators and induce inflammation and neurotoxicity, and alternative (M2), which release anti-inflammatory mediators and induce anti-inflammatory and neuroprotective effects, although there is a continuum of different intermediate phenotypes between M1 and M2, and microglia can transfer from one phenotype to another. M2 activation is induced by anti-inflammatory cytokines, such as interleukin (IL)-4, IL-13, and IL-10, by binding of Fc receptors to immune complexes, detection of apoptotic cells, and by activation of transcription factors [5]. M2 microglia release transforming growth factor-β (TGF-β), insulin-like growth factor-1 (IGF-1), fibroblast growth factor (FGF), nerve growth factor (NGF), brain-derived neurotrophic factor (BDNF), and glial cell-derived neurotrophic factor (GDNF), induce mannose receptor (CD206) and arginase 1 (Arg1), promote phagocytosis of cell debris and misfolded proteins, extracellular matrix reconstruction, and tissue repair, and support neuronal survival [6]. M1 activation can be induced by, for example, interferon-γ (IFN-γ) and lipopolysaccharide (LPS), and produces inflammatory cytokines and chemokines, such as tumor necrosis factor-alpha (TNF-α), IL-6, IL-1β, IL-12, and CC chemokine ligand (CCL) 2 [5].

Modulation microglia M1/M2 polarization and shifting from M1 to M2 phenotype have been suggested as promising therapeutic strategies in neurodegenerative diseases, such as Alzheimer’s (AD) and Parkinson’s (PD) disease, in which microglia-mediated neuroinflammation is a common feature. Some modulators were discovered that shifted microglia from pro-inflammatory M1 to anti-inflammatory M2 by inhibiting nuclear factor kappa-light-chain-enhancer of activated B cells (NF-κB) [6,7,8]. Furthermore, Zhang et al. have shown that microglia M1/M2 polarization may occur via mitogen-activated protein kinase (MAPK)-dependent inactivation [9]. Likewise, research on activator protein 1 (AP-1), signal transducer, and activator of transcription (STAT) transcription factors suggest that inhibiting them probably results in polarization from M1 to M2 microglia [6]. It was also shown that activation of the peroxisome proliferator-activated receptor gamma (PPARγ) pathway, in conditions of ischemia/stroke, regulates the balance of pro-/anti-inflammatory microglia polarization [10].

Although the vast majority of research is focused on microglia as key regulators of neuroinflammation [11,12,13], more data highlight the importance and contribution of astrocytes to the inflammation found in various CNS diseases, including neurodegenerative disorders such as AD and PD [14,15,16]. Under normal conditions, the main task of astrocytes is to maintain the proper activity of neurons by glial transmission, adult neurogenesis, neurotransmission, maintaining ionic homeostasis, protection against intoxication, participation in the metabolism, and storage of glucose and glycogen. Once exposed to damaging factors, astrocytes are activated and, similarly to microglia, show dual nature (Figure 2). Reactive astrocytes are induced by classically activated neuroinflammatory microglia via secretion of IL-1α, IL-1β, TNF-α, and complement component 1q (c1q), leading to a loss of their ability to promote neuronal survival, outgrowth, synaptogenesis, and phagocytosis, and inducing the death of neurons and oligodendrocytes [17].

Activated astrocytes are also able to create a glial scar isolating and protecting healthy tissue from the neurotoxic environment in parallel to attenuating regenerative processes of the injured tissue [18]. The signaling molecules involved in the induction of pro-inflammatory A1 astrocytes or inhibition of anti-inflammatory A2 astrocytes are NF-κB, signal transducer and activator of transcription 3 (STAT3), circlgf1r, Kir6.2, and microRNA2, while signaling molecules responsible for the inhibition of A1 astrocytes or induction of A2 astrocytes are phosphoinositide 3-kinase (PI3K)/protein kinase B (Akt), STAT3, tropomyosin receptor kinase B (TrkB), connexin30, chemokine receptor 7 (CXCR7), 17β-estradiol, FGF, milk fat globule epidermal growth factor 8 (MFG8), and TGF-β [19].

In the acute phase after injury, local neuroinflammation is tightly controlled, may have neuroprotective significance, and may be one of the mechanisms that increase compensatory processes activated in the damaged area of the CNS. In some circumstances, the balance of inflammatory and intrinsic repair processes that influence the functional recovery of CNS may be disturbed, leading to chronic inflammatory reactions, secretion of harmful factors, and propagation of the pathological process. A positive aspect of the interaction of the immune system with the nervous system is the release of neuroprotective mediators by glial cells, of which NTFs deserve special attention. NTFs can support the growth, survival, and differentiation of neurons. Additionally, the recent literature suggests that they may also be important regulators of inflammation [20] and mitochondrial function [21,22]. Since dysregulation of inflammatory pathways and mitochondria dysfunction are common features of CNS pathologies, especially in PD and AD, and it is suggested that NTFs can limit neuroinflammation, in this review we are aiming to outline the current state of knowledge about overlapping expression patterns of NTFs in the immune and nervous systems and the ability of NTFs to influence the activity of immune cells and mitochondria-dependent inflammation in the CNS.

## 2. Neurotrophic Factors in the Healthy and Diseased Brain

Stress is part of the daily life of individuals. Although short-term stress can be associated with positive changes in the brain, such as improving cognitive function, chronic stress is destructive to the nervous system, leading to microglia activation, the release of pro-inflammatory substances, recruitment of peripheral immune cells to the brain, and, thus, the initiation of an inflammatory response and an increase in the risk of developing neuropsychiatric disorders [23]. The proper transition from the pro-inflammatory to the anti-inflammatory phenotype of glial cells means that the pathology can be successfully repaired by the release of substances such as anti-inflammatory cytokines, growth factors (GFs), and NTFs, thereby possibly preventing cell death and promoting neuronal survival and neuroprotective processes. NTFs are a group of GFs that are active both in prenatal and adult life. In prenatal life, most neurons die, and the survival of individual cells depends on access to NTFs [24], while in adulthood, NTFs are involved in the processes responsible for maintaining the balance between neuroregenerative and neurodegenerative processes. Although NTFs are mainly synthesized and released in the CNS by neurons and glial cells [25,26], their expression is also observed in cells of the peripheral nervous system and other non-neuronal peripheral cells such as T and B lymphocytes, monocytes [27], vascular endothelial [28], and smooth [29] and skeletal muscle cells [30]. The first identified neurotrophic factor was NGF. Since the discovery of NGF, several other NTFs have been described, each with a unique character and biological activity. Depending on the type of specific neurotrophic factor activated and its receptor, signaling pathways for neuronal differentiation, maturation and survival, axonal and dendrite growth, synapse formation, or apoptosis can be triggered (Figure 3). Furthermore, NTFs are involved in the enhancement of synaptic plasticity, a key process for learning and memory.

Recent evidence suggests that alterations in NTFs, their dysregulated sorting and secretion, and the loss of their trophic support for selective neuronal populations may lead to the neuronal degeneration characteristic of AD, PD, and other neurodegenerative and neuropsychiatric diseases [31,32,33]. The etiopathology of neurodegenerative diseases is still not understood but it is known that chronic neuroinflammation and NTF dysregulation are processes involved in the initiation and/or progression of CNS diseases [34,35]. Similar to the pathomechanism of neurodegenerative diseases, data suggest that abnormal regulation of NTFs is also involved in the course of conditions related to impaired development of the brain and psychosis. Studies on various animal models indicate that even maternal immune activation (MIA) during pregnancy is a sufficient factor in developing neuropsychiatric disorders in the offspring, such as autism [36,37].

### 2.1. BDNF

BDNF is one of the most studied NTFs. BDNF controls synaptic plasticity through its learning and memory processes, which are interrelated phenomena. The action of BDNF and the type of activated signaling pathway depends on which biologically active isoforms bind to different types of its receptors. Because the precursor of BDNF (proBDNF) and mature BDNF (mBDNF) often exert an opposite effect on the survival, differentiation, growth, and apoptosis of neurons, their balance is an important factor in the regulation of many processes in the CNS. The mBDNF/TrkB receptor complex triggers signaling pathways responsible for dendritic growth, pine maturation and stabilization [38,39,40], development of synapses [41,42], learning- and memory-processes-dependent synaptic plasticity [43,44], and survival of neurons [45,46,47,48].

Howells et al. have examined the BDNF mRNA expression in substantia nigra pars compacta (SNpc) of PD patients and showed its 70% reduction compared to controls. Although this reduction partially resulted from the loss of dopaminergic neurons in SNpc, which express BDNF, surviving dopaminergic neurons also expressed less BDNF mRNA [49]. A Danish case-cohort study has revealed an association between lower blood levels of BDNF in newborns and an increased risk of developing autism spectrum disorder (ASD) [50]. Additionally, researchers found a trend toward elevated levels of inflammatory markers and reduced levels of BDNF [50]. On the other hand, a meta-analysis performed by Liu et al. showed higher levels in the peripheral blood of BDNF, NGF, and vascular endothelial growth factor (VEGF) in children with ASD compared to healthy controls [51]. Thus, BDNF expression seems to be delayed in ASD cases [50]. Interestingly, post-mortem examination of the brains of children with ASD showed an increased number of prefrontal neurons, which can be a result of abnormal regulation of BDNF, leading to subsequent excess of axonal connections, and disrupting the process of circuit formation [51,52,53]. Most previous studies have shown decreased peripheral levels of BDNF in schizophrenic patients. Furthermore, the lower peripheral level of BDNF in individuals with an at-risk mental state for psychosis compared to first-episode psychosis and chronic schizophrenia patients was investigated [54]. Thus, BDNF has been proposed to play a role as an indicator of the risk of psychosis development and cognitive deficit in schizophrenia [55].

### 2.2. GDNF

GDNF was first purified from rat B49 glial cell line and characterized by promoting the survival and morphological differentiation of dopaminergic neurons and increasing high-affinity dopamine uptake by them [56]. GDNF binds to the GDNF family receptor (GFR) α1 or, with lower affinity, to GFRα2 or GFRα3, and transmits signals by its receptor tyrosine kinase rearranged during transfection (RET). The biological responses of GDNF after RET activation and initiation of the signaling of the MAPK, PI3K/Akt, and proto-oncogene tyrosine-protein kinase (Src) pathways are morphological transformation, proliferation, cell migration, neurite elongation, and neurite branching [57]. Independently of RET signaling, GDNF-GFRα1 complex may bind neural cell adhesion molecule (NCAM) and activate Fyn and FAK protein kinases supporting Schwann cell migration and axonal growth in hippocampal and cortical neurons [58]. GDNF was shown to be expressed in many different regions of the developing nervous system. Levels of GDNF were observed in the thalamus, hippocampus, cerebellum, cortex, spinal marrow, and substantia nigra (SN) [59,60]. In addition, GFRα1 and GDNF were found to be expressed in the thymus, which may indicate the relationship between GDNF and the development and function of immune cells. It has been shown that GDNF protects dopaminergic neurons, increases the number of tyrosine hydroxylase (TH)-positive cells, and protects neurons from toxic assault [61,62,63,64].

Since many studies have demonstrated the crucial role of GDNF in maintaining the proper functioning of dopaminergic neurons, it has been proposed to reduce GDNF expression in the brains of PD patients. Chauhan et al. found that the level of GDNF was significantly reduced in the SNpc of patients with PD compared to the control group [65,66]. On the contrary, Hunot et al., using in situ hybridization, observed no detectable expression of GDNF, possibly due to very low levels of expression in the adult human brain [65].

### 2.3. CDNF

Cerebral dopamine neurotrophic factor (CDNF) is a novel neurotrophic factor that is structurally and mechanistically distinct from other growth factors. CDNF has recently been shown to have therapeutical properties in PD [67]. The mechanism of CDNF action is associated mainly with the regulation of endoplasmic reticulum (ER) function by modulation of unfolded protein response (UPR) pathways, which creates new research opportunities in the modulation of the inflammatory response [20]. Arancibia et al. have confirmed that CDNF triggers the induction of an adaptive UPR and increases binding immunoglobulin protein (BiP), activating transcription factor 4 (ATF4), transcription factor 6 (ATF6), and X-box binding protein 1 (XBP-1) expression both in HEK293-T cells and hippocampal neurons, thus leading to inhibition of ER-stress [68].

While the data concerning the changes in the expression of CDNF in various CNS disorders are scarce, interesting studies are showing that CDNF alleviates ER stress-induced cellular damage and suppresses the secretion of pro-inflammatory cytokines from astrocytes [69], thus promoting recovery and survival of midbrain dopaminergic neurons in animal models of PD [70,71]

### 2.4. NGF

NGF was discovered by Rita Levi-Montalcini and collaborators in the 1950s, and originally was characterized by its ability to stimulate growth, differentiation, survival, and maintenance of peripheral sympathetic neurons, sensory neurons, and cholinergic forebrain neurons in CNS [72,73]. NGF is a neurotrophin that is abundantly expressed in the CNS in neurons, oligodendrocytes, microglia, and astrocytes as well as in the periphery [73,74]. NGF acts through the p75^NTR^ and tropomyosin receptor kinase A (TrkA) receptors, often with the opposite effect. By acting on TrkA receptors, NGF triggers three main signaling pathways: Ras/Raf/MAPK, phospholipase C-γ (PLC-γ), and PI3/Akt kinase, and leads to neuronal survival, growth, and differentiation. While acting on the p75^NTR^ receptor, it stimulates NF-κB and MAPK- c-Jun N-terminal kinase (JNK) pathways, leading to neuronal survival and apoptosis, respectively [73]. NGF is essential for maintaining the phenotype of cholinergic neurons [75,76]. Administration of NGF into the brain prevents degeneration of damaged cholinergic neurons, increases the activity of undamaged cholinergic neurons, and affects spatial memory and recognition in aged rats. NGF was shown to be upregulated in inflammation [77,78] and chronic pain [79,80].

Several studies have reported the preventive role of NGF in the course of AD [81,82,83]. Peng et al. have shown a negative correlation between proNGF levels and Mini-Mental Status Examination (MMSE) score, demonstrating that the accumulation of proNGF correlated with loss of cognitive function [84], whereas biochemical and immunocytochemical studies of TrkA, TrkB, and TrkC levels demonstrated their reduction in the nucleus basalis of Meynert in AD patients [85]. Scott et al., using two-site ELISA, have shown upregulated NGF levels in the hippocampus, superior temporal gyrus, superior frontal gyrus, inferior parietal lobule, frontal and occipital cortical poles, cerebellum, amygdala, and putamen compared to age-matched control and PD cases. Only in the nucleus basalis of Meynert the NGF level was significantly reduced. This NGF upregulation may be due to the widespread distribution of TrkA throughout the brain, and abnormalities in the utilization, internalization, or transport of NGF in the course of AD could play a role in the widespread increase in NGF-like activity [86].

Data presented above suggest that each CNS disease may have a unique profile of NTFs and, upon looking more deeply into molecular pathways of abnormally regulated NTFs, the pathogenesis of some diseases may be partially explained (Table 1).

## 3. Neurotrophic Factors in Glia-Neuronal Crosstalk and Their Role in Neuroinflammation

Neurotrophic factors are crucial for proper CNS structure and function and play a central role in the regulation of signaling between microglia or astrocytes and neurons. Firstly, the relationship between NTFs and glia affects the expression of inflammatory mediators (Table 2). However, NTFs, such as GDNF and BDNF, released from glial cells also play a significant role in glia-mediated synapse formation, since pre- and postsynaptic terminals cooperate with astrocytes and microglia in “quad-partite” synapses [90]. BDNF expressed and secreted by glial cells promotes the development of inhibitory synapses [91,92]; similarly, GDNF promotes the formation of new synaptic terminals and increases dopamine release through an unknown mechanism [93]. In addition, astrocytes can prevent excitotoxicity by releasing GDNF and NGF, which support neuronal survival [94], suggesting that glia-derived neurotrophic factors could play significant roles during neurodegenerative disorders. What is more, some exogenously administered NTFs, such as BDNF and NGF, affect glial activation states with beneficial effects on these disorders’ outcomes. Jiang et al. have reported that intranasal administration of BDNF modulated the local inflammatory process, in the rat’s brain after a stroke, on the cellular, cytokine, and transcription level, through activation of anti-inflammatory microglia, downregulation of both protein and mRNA levels of TNF-α, upregulation IL-10 and mRNA levels, and modulation of NF-κB activity [95]. Furthermore, it was shown that BDNF pretreatment suppressed the expression of inflammatory factors, including TNF-α, IL-1β, and IL-6, and increased the expression of the anti-inflammatory factor IL-10 in a rat model of *Streptococcus pneumoniae* meningitis. In vitro data have shown that overexpression of CDNF in astrocytes reduced secretion of inflammatory cytokines induced by ER stress [69], whereas microglia suppressed LPS-induced release of pro-inflammatory cytokines [96]. The anti-inflammatory properties of NTFs were also reported by Rickert et al. [97]. Their study revealed that pretreatment of primary rat microglia with factor GDNF family members reduced the LPS-induced expression of pro-inflammatory cytokines and cyclooxygenase-2 (COX-2) [97]. Furthermore, the ability of GDNF to suppress the activation of microglia was also confirmed in animal models of PD [96,98]. Another key mediator in crosstalk between the nervous and immune systems is NGF. It has been demonstrated that NGF acts on microglia, directing them toward a neuroprotective and anti-inflammatory phenotype, increases amyloid beta (Aβ) uptake by microglia, and enhances its degradation [99].

### 3.1. The Role of BDNF in Neuroinflammation

Since BDNF downregulation and neuroinflammation play a crucial role in the pathogenesis of many brain disorders, the question arises whether BDNF can affect inflammatory-related processes and reduce pro-inflammatory progress. However, there is still little research into the mechanism underlying the anti-inflammatory properties of BDNF. Liang et al. have demonstrated that overexpression of BDNF in a spinal cord injury model induced the expression of TrkB and suppressed the p38-MAPK signaling pathway, thus reducing inflammation, as indicated by decreased levels of TNF-α, IL-1β, IL-6, IL-18, inducible nitric oxide synthase (iNOS), and COX-2 [100]. Inverse results were obtained when TrkB ANA-12 inhibitor was used. One of the key processes induced in response to BDNF action by the stimulation of PI3K/Akt seems to be the modulation of NF-κB, which is a pivotal mediator of chronic stress and inflammatory responses. An additional mechanism of anti-inflammatory properties of BDNF is the reduction of the activity of glycogen synthase kinase 3 beta (GSK-3β) after the stimulation of PI3K/Akt and extracellular signal-regulated kinase (ERK) 1/2 pathways [102,103,104,105]. The inhibition of GSK-3β prevented the phosphorylation of NF-κB, diminished the production of pro-inflammatory cytokines [111], and increased cAMP response element-binding protein (CREB) DNA binding activity [112]. Wu et al. have revealed the ability of BDNF to inhibit microglial activation due to an upregulation of MAPK-1 and subsequent dephosphorylation of p38 and JNK, and its ability to activate the Erk-CREB pathway [113]. It has been demonstrated that activated CREB could inhibit NF-κB activity through competition for limited amounts of CREB-binding protein, an important coactivator in regulating the transcriptional activity of these factors [113,114] Interestingly, it was found that the BDNF affecting PI3K/Akt and ERK signaling pathways induces nuclear factor erythroid 2-related factor 2 (Nrf2) activation [106,115], which is mainly known as a regulator of cellular antioxidant defense but has also been shown to be involved in anti-inflammatory pathways by interacting with NF-κB signaling. In Nrf2-mediated NF-κB inhibition, heme oxygenase-1 (HO-1) plays an important role, inhibiting NF-κB-mediated transcription of adhesion molecules, such as E-Selectin and vascular cell adhesion molecule 1 (VCAM-1), in endothelial cells [116]. Understanding how neuroinflammation is involved in CNS disorders and what role BDNF plays in neuroinflammation may be critical to the development of therapeutic strategies. The interaction between BDNF and neuroinflammation is strongly related to the interaction with NF-κB. However, the exact mechanism of this interaction is not well understood, so there is a need for more research into the anti-inflammatory properties of BDNF.

### 3.2. The Role of GDNF in Neuroinflammation

Since chronic neuroinflammation plays a key role in the pathomechanism of many neurodegenerative diseases, such as AD and PD, and in neurodevelopmental disorders such as autism, compounds that have a modulating effect on this harmful process are wanted. Recent studies indicate that GDNF may have anti-inflammatory properties. Rocha and colleagues have found that GDNF derived from astrocytes was able to inhibit rat midbrain microglial activation induced by the TLR2 agonist Zymosan A by stimulating the intracellular signaling cascade GFRα1–Ret [57]. Furthermore, Chou et al. have shown therapeutic effects of adenoviral-mediated GDNF on neuropathic pain behaviors in rats by inhibiting microglia activation and cytokine production via modulation of p38 and protein kinase C (PKC)/iNOS signaling [101]. Bilateral intra-striatal administration of GDNF in Gdnf^+/−^ mice normalized several parameters including locomotor activity, GDNF protein levels within the striatum and SN, the number of TH-positive neurons in the SN, upregulation of COX-2, and downregulation of superoxide dismutase (SOD)-2 protein levels in the SN. Rickert et al. have demonstrated that GDNF can reduce the production of microglial NO and mRNA levels of IL-1β, TNF-α, IL-6, and COX-2 by reducing phosphorylation of p38, which could result in subsequent neuroprotection [97]. Different microenvironments in neurological pathologies, and especially in in vitro conditions, may lead to different forms of microglial activation. Experiments performed by Wang et al. have indicated that in subacute cerebral ischemia, activated microglia exert a neuroprotective role through balancing the expressions of GDNF and TNF-α, whereas the inhibition of microglial activation by poly [ADP-ribose] polymerase 1 (PARP-1), the co-activating factor of nuclear factor NF-κB signaling pathway, attenuates GDNF production [117]. Researchers speculate that inhibition of microglial activation attenuates the phosphorylation level of ERK, leading to reduced GDNF secretion [117]. Using a 6-hydroxydopamine (6-OHDA) rat model of PD, Wang et al. have demonstrated that lipid-coated GDNF microspheres injected into the striatum and stimulated with low-frequency ultrasound reduced apomorphine-induced rotations, increased striatal dopamine and nigral TH levels, and reduced caspase-3, TNF-α, matrix metalloproteinase 9 (MMP-9), and MHC II compound levels induced by 6-OHDA [118]. Qing et al. have demonstrated the role of the Hippo/ yes-associated protein (YAP) pathway in the anti-inflammatory effect of GDNF against Aβ [110]. The Hippo/YAP signaling pathway was shown to be involved in the renewal of neural stem cells, the proliferation of neural progenitor cells, differentiation and activation of glial cells, and myelination by glial cells as well as in the development of inflammatory-related diseases [119]. Qing’s group revealed that treatment with GDNF upregulated YAP expression and reduced the production of TNF-α, TGF-β, IL-1β, and IL-12β, in a dose-dependent manner, whereas YAP knockdown lowered the function of GDNF in microglial cells. There is some indication that modulating microglial polarization may be a suitable strategy to treat neuroinflammation. Zhong et al. have shown the ability of GDNF produced by adipose-derived stem cells (ADSCs) to inhibit the microglia M1 phenotype, reduce the release of inflammatory TNF-α and iNOS, and promote the M2 phenotype by upregulating the PI3K/Akt pathway and then increasing the production of anti-inflammatory IL-10, IL-4, and TGF-β1 [120]. Collectively, these data further support the important role GDNF plays in maintaining the normal activation state of microglia and protecting neurons, especially dopaminergic ones.

### 3.3. The Role of CDNF in Neuroinflammation

CDNF has demonstrated protective and restorative in neuropathology associated with ER stress and was shown to be able to suppress inflammation and apoptosis [107]. Zhao et al. have provided evidence that CDNF reveals the anti-inflammatory properties by inhibition of JNK but not the p38 or ERK signaling pathways in LPS-induced microglia [96]. Furthermore, in further studies, authors have indicated the involvement of CDNF in the reduction of LPS-induced inflammation through the competitive inhibition of Akt phosphorylation and suppression of downstream pathways, including forkhead box protein O1 (FoxO1) and mTOR signaling [105]. Another study showed that the modulation of inflammatory responses caused by CDNF is not only by decreasing microglia/macrophage recruitment but also regulating its polarization and enhancing M2 subset polarization [121]. Degeneration of dopaminergic neurons can be caused by several processes, among which the oligomerization of α-synuclein and the deposition of these forms of α-synuclein in the cytoplasm of neurons play a key role. Albert et al. have identified a direct inhibitory effect of CDNF on α-synuclein aggregation, mediated by its high-affinity association with α-synuclein monomers. CDNF application reduced preformed fibrils (PFFs) uptake by approximately 25%, and CDNF had a stronger effect on oligomeric species as opposed to the fibrillar form [109]. The inhibitory effect of CDNF on α-synuclein aggregation may be due to ER stress modulation and UPR modulation or binding to α-synuclein or reduction of α-synuclein uptake by its receptors. Even though phosphoSer129-α-synuclein inclusions were detected in the rodent brains after PFFs injection into the striatum, significant loss of TH was not observed and the effects of CDNF on the number of phosphoSer129-α-synuclein inclusions were not indicated in the SN. CDNF likely affects the localization and the kinetics of the association of α-synuclein to the aggregates but not its total levels [109]. Lindahl et al. also did not observe a change in the number of dopaminergic neurons in SN and the concentration of dopamine and its metabolite in the striatum in Cdnf^−/−^ mice; however, there was an age-dependent deficit in the function of the dopamine system in Cdnf^−/−^ male mice observed as D-amphetamine-induced hyperactivity, aberrant dopamine transporter function, and as increased D-amphetamine-induced dopamine release [122]. Although CDNF receptors and their signaling pathways are still poorly understood, there are more and more reports of its neuroprotective and neurorestorative effects on TH-positive cells in the nigrostriatal dopaminergic system and its inhibitory effect on the synthesis and release of pro-inflammatory cytokines that reduce neuroinflammation.

### 3.4. The Role of NGF in Neuroinflammation

Since a significant increase in NGF synthesis has been reported in a wide range of inflammatory conditions [78,123,124], NGF is considered to be one of the mediators in crosstalk between the nervous and immune systems. The way by which NGF modulates the immune response is however still not fully understood. NGF is known to support the growth of cholinergic neurons [125,126] and is also associated with embryonal development and the differentiation of peripheral neuronal cells [127]. Tuszynski has shown that intraparenchymal NGF infusions in rats prevented the degeneration of basal forebrain cholinergic neurons compared to vehicle-infused animals and cholinergic axons sprouted toward the NGF infusion in an apparent gradient-dependent manner [128]. It was demonstrated that brain cholinergic signaling may regulate local brain inflammation. Terrando et al. have revealed the effect of trauma and endotoxemia on brain function in the context of cholinergic signaling [129]. Administration of a selective α7 subtype nicotinic acetylcholine receptor (α7 nAChR) agonist after LPS significantly improved neuroinflammation and hippocampal-dependent memory dysfunction in mice. The mechanism underlying this process was likely the modulation of NF-κB activation in monocytes and regulation of the oxidative stress response through nicotinamide adenine dinucleotide phosphate (NADPH) signaling. NGF increases the density of innervation, sprouting of axonal endings, and dendritic arborization of neurons, and in the conditions of the inflammatory process it can lead to indirect regulation of the immune response, but also the regulation of the production of neurotransmitters and neuropeptides that have a direct impact on peripheral immune cell [78]. Prencipe et al. have reported an increase in TrkA expression in monocytes after TLRs stimulation. NGF, via its high-affinity receptor TrkA exerted an anti-inflammatory effect by increasing Akt phosphorylation, inhibiting GSK-3β activity, reducing IκB phosphorylation and p65 NF-κB translocation, and increase of nuclear p50 NF-κB binding activity [130]. Furthermore, a study performed by Fodelianaki et al. revealed that NGF binding to TrkA downregulated LPS-induced production of pro-inflammatory cytokines and NO in primary mouse microglia and inhibited TLR4—mediated activation of the NF-κB and JNK pathways [108]. Therefore, NGF/TrkA activation may be one of the signals that, when inflammatory stimuli act and the activation of the immune system occurs, is involved in the regulation of the balance between pro- and anti-inflammatory pathways.

## 4. Possible Interaction between Neurotrophic Factors, Mitochondria, and Neuroinflammation

The pathomechanism of neurodegenerative disorders such as AD and PD is morphologically distinct and still only partially understood; however, it is a component of several general pathological features, such as mitochondrial dysfunction, oxidative stress, and associated neuroinflammation. Loss of trophic support for neurons also plays a key role in neurodegeneration. For this reason, it was hypothesized that NTFs may control the course of inflammation by regulating the function and energy capacity of mitochondria. Mitochondria are targets of stress by many factors, including aging, pathogens, misfolded and/or aggregated proteins, such as α-synuclein, and environmental factors. Mitochondrial stress results in the release of DAMPs that activate innate immune receptors and downstream signaling. Chronic activation of specific pathways results in mitochondria-dependent inflammation (mitoflammation) and associated pathology. Growing evidence from animal and in vitro studies have suggested that mitochondrial function and NTFs, particularly BDNF, have a complex and reciprocal relationship, and that NTFs and mitochondria can modulate each other’s functions. It was shown that the neurotrophic receptor TrkB is embedded in the mitochondrial membranes [131], which supports the thesis that neurotrophins may modify mitochondrial functions. Recent data have described that NTFs may affect cytoskeletal rearrangements and axonal branching, influence the motility and actin-based docking of mitochondria in axons, induce an initial burst of fission of mitochondria, or affect mitochondrial integrity by coupling of the oxidation to adenosine triphosphate (ATP) synthesis [132,133,134]. Mitochondria act as intracellular Ca^2+^ buffers, but when mitochondria are overloaded with Ca^2+^, there is an increase in ROS production, ATP synthesis inhibition, and mitochondrial permeability transition pore (mPTP) opening, leading to the release of proapoptotic proteins and the processes of necrotic and apoptotic cell death. Although ROS at physiological concentrations plays a pivotal role in several signaling pathways, such as cell cycle regulation, phagocytosis, and enzyme activation, excessive generation of ROS leads to several harmful effects, including mitochondrial damage, the release of mitochondrial DNA (mtDNA), and, in turn, chronic mitoflammation and disease. Many studies have been conducted to explore the mechanism by which mtDNA is released into the cytoplasm space, which can cause cellular stress. Release of mtDNA following mitochondrial outer membrane permeabilization (MOMP) in a BAX/BAK-dependent manner allows the extrusion of newly unstructured inner membrane, facilitating further mtDNA release into the cytoplasm, activating cyclic GMP–AMP synthase (cGAS) and generating cyclic guanosine monophosphate–adenosine monophosphate (cGAMP), which binds directly to the stimulator of interferon genes (STING) pathway and further recruits and activates tank-binding kinase 1 (TBK1) [135,136]. TBK1 induces the expression of various interferon (IFN)-stimulated genes and activates the NF-κB signaling pathway through phosphorylation, thus increasing the expression of IL-6 and TNF-α [135,137]. Furthermore, the accumulation of mtDNA in the cytoplasm may spread to the extracellular space and act on nearby microglia and astrocytes [138]. ROS formation is tightly associated with endothelial cell dysfunction, and BDNF has been shown to possess a microvascular protective effect. BDNF secreted by brain microvascular endothelial cells was observed to suppress mitochondrial swelling, inhibit oxidative stress evaluated by intracellular ROS formation, SOD activity, malondialdehyde, and NO content, prevent early apoptosis under hyperglycemic conditions, and induce mitophagy through the hypoxia-inducible factor 1/BCL2 interacting protein 3 (HIF-1α/BNIP3) signaling pathway [139]. It has been also shown that BDNF induces the nuclear translocation of Nrf2, thus protecting cells against the oxidative damage caused by injury and inflammation. Bruna et al. have revealed that BDNF-induced Nrf2 nuclear translocation requires ROS and ryanodine receptor-mediated Ca^2+^ signals, and the participation of the classical ERK and PI3K signaling pathways [106]. In addition, BDNF controls mitochondrial transport in neurons and affects mitochondria function through the regulation of the respiratory control index (RCI) [140]. This BDNF-evoked increase in the efficiency of respiratory coupling, ATP synthesis, and organelle integrity has important implications for therapeutic approaches in neurodegenerative diseases [22]. Another research group has shown that depriving sympathetic neurons in cell culture of NGF led to a Bax-dependent increase of mitochondrial-derived ROS in response to leakage of electrons from the mitochondrial electron transport chain. When NGF was re-added to these cells, glutathione redox cycling was activated, reducing H_2_O_2_ levels and blocking the release of cytochrome c [141,142]. Sun et al. have demonstrated that NGF reduces the production of ROS in the primary cortical neurons exposed to oxygen-glucose deprivation through a higher expression of HO-1 controlled by the MAPK/ERK survival pathway [143]. There is still a gap in research into the direct effects of NTFs on mitochondrial function in the context of neuroinflammation. The abovementioned studies indicate, however, that the link between the action of NTFs and the alleviation of neuroinflammation is their inhibitory effect on oxidative stress, including the level of mitochondria-derived ROS production.

## 5. Conclusions and Further Prospects

Glial cell activation and neuroinflammation, mitochondrial dysfunction, and dysregulation of NTFs contribute to the pathomechanism of various diseases of the CNS. There is a growing interest in the interaction between the nervous and the immune system and the impact of this relationship on homeostasis or, in cases when pathological stimuli act, on the development of inflammatory diseases. Thus, the normalization of glial dysfunction or the upregulation of neuroprotective ability may prevent progressive neurodegeneration. Data from the literature show that NTFs are mediators that can regulate neuronal cell function, immune cell activity, and oxidative stress. NTFs exert various effects and, depending on the receptor on which they act, may have a pro-inflammatory effect activating immune responses or anti-inflammatory properties activating the signaling cascades necessary to abolish inflammatory response and limit tissue damage. This review summarizes the ability of NTFs to influence the activity of immune cells. The proposed mechanism for this neuroimmune crosstalk may be due to the overlapping pattern of NTF expression in the immune and nervous systems.

Based on the promising results of animal studies [64,101,109,113,121,131], it was hypothesized that NTFs could be a potential regenerative tool in inflammatory and degenerative conditions of the CNS. Each NFT has its unique profile of action on specific subsets of neurons to ensure their optimal function. Considering this, GDNF and CDNF could be of particular interest for PD, NGF could be of particular interest for AD, and BDNF, because of its rich expression in the brain, could provide support for AD, PD, spinal cord injury, multiple sclerosis (MS), and Huntington’s disease (HD). Therefore, some NTFs have been investigated as potential therapeutic therapies in preclinical and/or clinical trials. Delivering NTFs to degenerating neurons appears to be a powerful neuroregenerative agent; however, the greatest challenge is found in determining how they should be delivered to their destination in the brain. Studies on the possibility of NTFs crossing the blood–brain barrier (BBB) are not clear. Some researchers state that some NTFs cross the BBB [144,145] and others do not [146]. Furthermore, there are no such studies on all NTFs. For the delivery of peripherally administered NTFs to the CNS, strategies for their chemical modification and/or conjugation are promising [147].

The first clinical trial with NTFs was the systemic administration of ciliary neurotrophic factor (CNTF) in amyotrophic lateral sclerosis (ALS) patients. Problems of CNTF with crossing the BBB and side effects terminated the trial [148,149,150,151,152]. Then, GDNF was directly injected into the cerebral ventricles of PD patients; however, side effects and no beneficial effects were seen [153]. The clinical phase I safety trial of direct intraputamenal GDNF infusion in patients with PD performed by Patel showed significant symptomatic improvement [154]; however, another randomized, controlled, and blinded clinical trial did not confirm these results [155]. In the next step, to solve problems with distributions of NTFs in the brain, Bachoud-Le Âvi et al. [156] and Aebischer et al. [157] introduced a macro-encapsulation technique, in which CTNF was administered stereotactically into the lateral ventricle of patients with HD and ALS. No toxicity, but also no clinical benefits were observed. Tuszynski et al. successfully performed a phase I trial of ex vivo NGF gene delivery in eight individuals with mild AD, implanting autologous fibroblasts genetically modified to express human NGF into the forebrain. Cognitive improvements and increases in cortical 18-fluorodeoxyglucose after treatment in serial PET scans were observed. Furthermore, a brain autopsy from one subject confirmed NGF expression in the cell grafts and robust growth responses to NGF [158]. Tuszynski et al. conducted another phase I clinical trial in which 10 patients with early AD underwent NGF gene therapy using ex vivo or in vivo gene transfer, and post-mortem analysis on 10 subjects with survival times ranging from 1 to 10 years post-treatment was performed. NGF gene therapy was safe and resulted in NGF-related axonal sprouting in all patients. This study also showed that therapeutic genes and activation of cell signaling were indicated in neurons with or without tau pathology [159]. Recently, more and more attention has been paid to the possibility of using CDNF in the therapy of PD. In a 2020 safety and efficacy study of intracerebrally administered CDNF protein therapy in patients with PD, use of a neurosurgically implanted drug delivery system (DDS) was completed; however, results were not made available [160]. The efficacy of some clinical trials indicates the powerful potential of NTFs in the therapy of CNS disorders. The limitations of NTF therapies are related to problems with the pharmacokinetics of NTFs and their proper delivery to the target site in the brain, rather than to their dubious neuroregenerative effects.

There is still a need for further research that would explore the relationship between the complex etiology of CNS diseases, NTF levels, and their interaction with immune cells. Understanding how NTF signaling pathways affect some molecular processes and the mechanism by which NTFs induce neuroprotective and anti-inflammatory effects can help against CNS damage and shed new light on the treatment of CNS disorders.

## Figures and Tables

**Figure 1 ijms-24-06321-f001:**
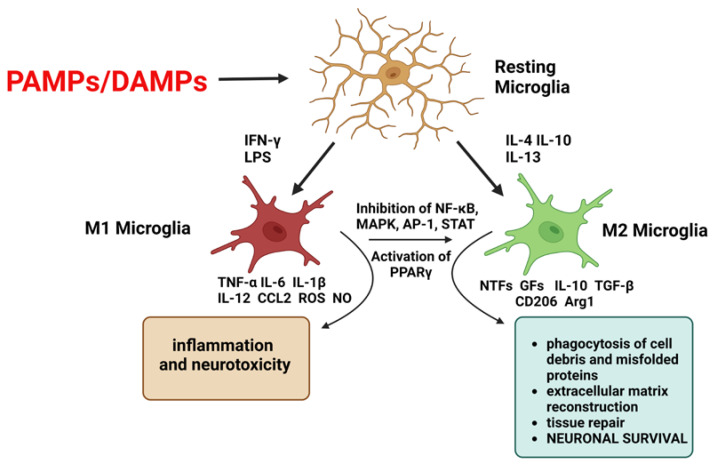
Microglia are activated by pathogen-associated molecular pattern molecules (PAMPs) and/or damage-associated molecular pattern molecules (DAMPs). The classical M1 activation can be induced by interferon-γ (IFN-γ) and lipopolysaccharide (LPS). M1 microglia produce inflammatory cytokines and chemokines, such as tumor necrosis factor-alpha (TNF-α), interleukin (IL)-6, IL-1β, IL-12, and CC chemokine ligand (CCL) 2, and induce inflammation and neurotoxicity. The alternative M2 activation is induced by anti-inflammatory cytokines such as IL-4, IL-13, and IL-10. M2 microglia release growth factors (GFs), and neurotrophic factors (NTFs), induce mannose receptor (CD206) and arginase 1 (Arg1), and promote neuronal survival. Shifting from M1 to M2 phenotype may occur via inhibition of nuclear factor kappa-light-chain-enhancer of activated B cells (NF-κB), mitogen-activated protein kinase (MAPK), activator protein 1 (AP-1), and signal transducer and activator of transcription (STAT) transcription factors, and activation of the peroxisome proliferator-activated receptor gamma (PPARγ) pathway. The figure was created with BioRender.com.

**Figure 2 ijms-24-06321-f002:**
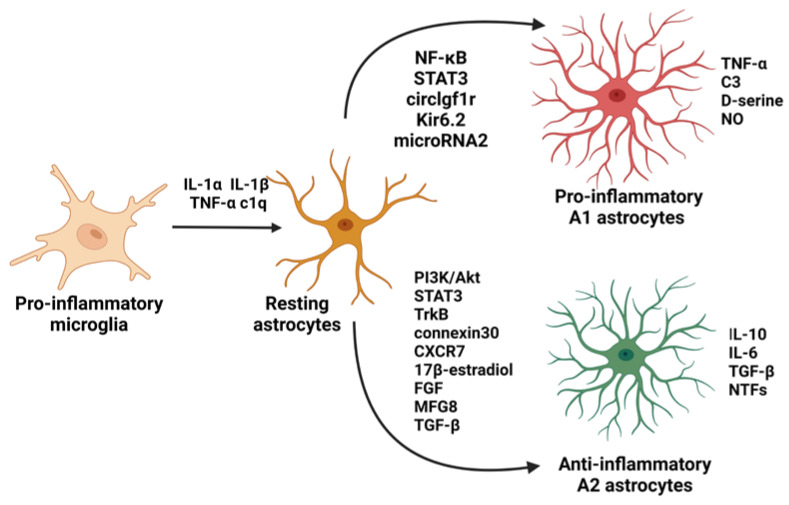
Reactive astrocytes are induced by classically activated neuroinflammatory microglia via secretion of interleukin (IL)-1α, IL-1β, tumor necrosis factor-alpha (TNF-α), and complement component 1q (c1q). Astrocytes show dual nature. The signaling molecules involved in the induction of pro-inflammatory A1 astrocytes are nuclear factor kappa-light-chain-enhancer of activated B cells (NF-κB), signal transducer and activator of transcription 3 (STAT3), circlgf1r, Kir6.2, and microRNA2, while signaling molecules responsible for the induction of A2 astrocytes are phosphoinositide 3-kinase (PI3K)/protein kinase B (Akt), STAT3, tropomyosin receptor kinase B (TrkB), connexin30, chemokine receptor 7 (CXCR7), 17β-estradiol, fibroblast growth factor (FGF), milk fat globule epidermal growth factor 8 (MFG8), and transforming growth factor-β (TGF-β). A1 astrocytes release interleukin (IL)-1β, TNF-α, and C3 components to propagate the neuroinflammatory response. They also release D-serine and nitric oxide (NO), which may contribute to excitotoxicity. A2 astrocytes appear to release anti-inflammatory compounds, such as neurotrophic factors (NTFs), IL-10, IL-6, and TGF-β, and promote the survival and growth of neurons and reparative functions. The figure was created with BioRender.com.

**Figure 3 ijms-24-06321-f003:**
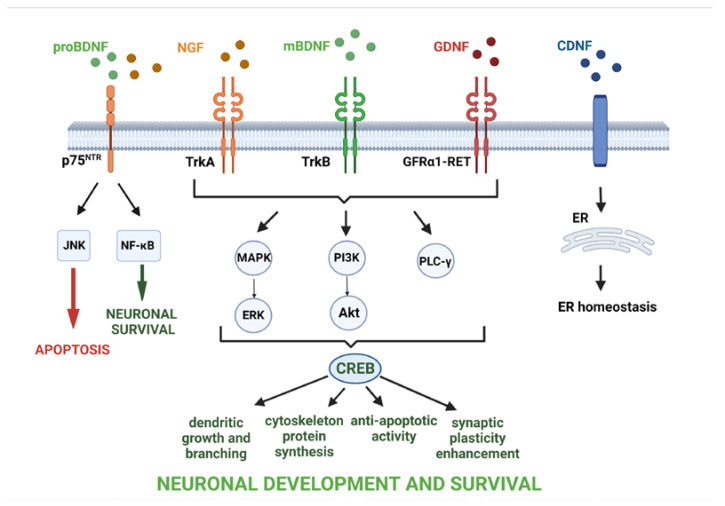
Signaling cascades activated by neurotrophic factors (NTFs). The precursor of brain-derived neurotrophic factor (proBDNF) or nerve growth factor (NGF) binding the p75^NTR^ leads to activation of c-Jun N-terminal kinase (JNK) or nuclear factor kappa-light-chain-enhancer of activated B cells (NF-ĸB) signaling pathways, which promote processes such as apoptosis and neuronal survival. The mature brain-derived neurotrophic factor/tropomyosin receptor kinase B (mBDNF/TrkB), NGF/tropomyosin receptor kinase A (TrkA), and glial cell-derived neurotrophic factor/GDNF family receptor α1/ receptor tyrosine kinase rearranged during transfection (GDNF/GFRα1-RET) complexes trigger activation of mitogen-activated protein kinase (MAPK), phosphoinositide 3-kinase (PI3K)/protein kinase B (Akt), and phospholipase C-γ (PLC-γ) pathways that, in turn, activate the cAMP response element-binding protein (CREB) and transcription of genes responsible for development and survival of neurons. Cerebral dopamine neurotrophic factor (CDNF) receptors and CDNF-activated signaling pathways are still poorly understood, but the mechanism of CDNF action is associated mainly with the regulation of endoplasmic reticulum (ER) function. The figure was created with BioRender.com.

**Table 1 ijms-24-06321-t001:** Neurotrophic factors (NTFs) expression in central nervous system (CNS) disorders and disease models.

NTF	Physiological Function	Disease	Model	Structure	Cell Type/Expression	References
Brain-derived neurotrophic factor (BDNF)	Binds to tropomyosin receptor kinase B (TrkB); activates phospholipase C-γ (PLCγ) and Ras; induces dendritic growth, development of synapses, survival of neurons; regulates synaptic plasticity	Parkinson’s disease (PD)	PD patients; post-mortem	Substantia nigra pars compacta (SNpc)	Dopaminergic neurons ↓	[49]
		Alzheimer’s disease (AD)	APP23 transgenic mice	Cortex	Astrocytes, microglia ↑	[87]
		Autism spectrum disorder (ASD)	Newborns	Peripheral blood	Dried blood spot samples ↓	[50]
			Children	Peripheral blood	↑	[51]
Glial cell-derived neurotrophic factor (GDNF)	Binds to GDNF family receptor (GFR)α1, GFRα2, and GFRα3; GDNF/ receptor tyrosine kinase rearranged during transfection (RET) activates mitogen-activated protein kinase (MAPK), phosphoinositide 3-kinase (PI3K)/protein kinase B (Akt), and proto-oncogene tyrosine-protein kinase (Src) pathways; GDNF/ neural cell adhesion molecule (NCAM) activates Fyn and FAK protein kinases, stimulates proliferation, cell migration, neurite elongation and neurite branching, Schwann cell migration, and axonal growth	PD	PD patients; post-mortem	SNpc	Neurons	[66]
				Mesencephalon/striatum	0	[65]
			1-methyl-4-phenyl-1,2,3,6-tetrahydropyridine (MPTP) mice model	Striatum	Parvalbumin-positive (PV+) interneurons ↔	[88]
Nerve growth factor (NGF)	Binds to tropomyosin receptor kinase A (TrkA); activates MAPK, PLC-γ, and PI3/Akt kinase; stimulates neuronal survival, growth, and differentiation; binds to p75^NTR^; stimulates c-Jun N-terminal kinase (JNK) and nuclear factor kappa-light-chain-enhancer of activated B cells (NF-ĸB); activates neuronal survival and apoptosis	AD	AD patients; post-mortem	The hippocampus, superior temporal gyrus, superior frontal gyrus, inferior parietal lobule, frontal and occipital cortical poles, cerebellum, amygdala, and putamen	*↑*	[86]
				Nucleus basalis of Meynert	↓	
			Mild cognitive impairment (MCI) and AD patients; post-mortem	Parietal cortex	*↑*	[85]
		PD	6-hydroxydopamine (6-OHDA) rat model	Striatum	Astrocytes ↑	[89]
		ASD	Children	Peripheral blood	↑	[51]

↑, expression is upregulated; ↓, expression is downregulated; ↔, expression is not changed; 0, no expression detected.

**Table 2 ijms-24-06321-t002:** Main signaling pathways involved in neurotrophic factors (NTFs) and glia-neuronal crosstalk.

Signaling Pathway	NTFs	Effect	References
Suppression of p38	Brain-derived neurotrophic factor (BDNF); Glial cell-derived neurotrophic factor (GDNF)	Reduction of inflammation (inhibition of production of inflammatory cytokines and enzymes)	[41,100,101]
Stimulation of phosphoinositide 3-kinase (PI3K)/protein kinase B (Akt), and extracellular signal-regulated kinase (ERK)	Nerve growth factor (NGF); BDNF; GDNF	Reduction of inflammation, upregulation of antioxidant defense, enhancement of neuronal survival (inhibition of glycogen synthase kinase 3 (GSK3) activity, suppression of nuclear factor kappa-light-chain-enhancer of activated B cells (NFĸB)-dependent transcription of pro-inflammatory cytokine genes, induction of nuclear factor erythroid 2-related factor 2 (Nrf2), increased cAMP response element-binding protein (CREB) activity)	[102,103,104,105,106]
Inhibition of c-Jun N-terminal kinase (JNK)	Cerebral dopamine neurotrophic factor (CDNF); BDNF	Regulation of microglia activation	[106,107,108]
Unknown	CDNF	Inhibition of α-synuclein aggregation	[109]
Stimulation of Hippo/yes-associated protein (YAP)	GDNF	Reduction of amyloid beta (Aβ)-induced inflammation	[110]

## Data Availability

Not applicable.

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
