# Peer review of "Glia-Neurotrophic Factor Relationships: Possible Role in Pathobiology of Neuroinflammation-Related Brain Disorders"

_ijms, 2023, doi:10.3390/ijms24076321_

Round 1

Reviewer 1 Report

Palasz et al summarized current understanding of glia and neurotrophic factors in the healthy state as well as pathological states, especially in neuroinflammatory and neurodegenerative diseases. Gliosis has been recognized and reported in various of clinical studies, animal models and in vitro cell culture systems used to study neurodegenerative diseases, however, targeting gliosis whether or not provides therapeutic effects in these models is still under debate. This manuscript reviewed recent literature from a novel perspective, by adding neurotrophic factors and interactions with glia, which may open up new directions for future researchers. However, there are several points that need more attention and improvements listed as follows.

1.      The authors put ‘neurotrophic factor and glia crosstalk’ in the title, which is not fully and clearly reflected in the main text, with the ‘crosstalk’ between glia or neuron not clearly stated.

2.      Each section contains a lot of text with specific information regarding each signaling pathways, molecules under different disease states. It would be easier for the readers to follow and more helpful find the most relevant info if the authors could make some changes to reorganize the paragraphs. For example, divide into subsections based on markers, diseases or cell types.

3.      The authors provided 2 figures that are very helpful for the readers to get an overview of how astrocytes and microglia function and get polarized upon different stimuli. It would be more helpful if they can add another diagram to show how neurotrophic factors get into the interactions, especially how all of them are integrated together to affect neuron and neuronal functions.

4.      Adding a few tables summarizing key findings and references may also help the readers navigate through the manuscript.

Reviewer 2 Report

The papers gave an overview of the current state of knowledge about the processes affected by NTFs, the crosstalk between NTFs, mitochondria, and the nervous and immune systems, leading to the inhibition of neuroinflammation and oxidative stress, underlying various CNS disorders.

The review is interesting, well written, and organized. I have only minor suggestions.

In the chapter Conclusions and further prospects, I suggest authors write in more detail about future research and approaches, including potential clinical studies that will contribute to the better understanding of complex mechanisms of NTFs' neuroprotective and anti-inflammatory effects in CNS disorders.

Minor: References 6 and 7 are the same
